# Proteomic Biomarkers of the Apnea Hypopnea Index and Obstructive Sleep Apnea: Insights into the Pathophysiology of Presence, Severity, and Treatment Response

**DOI:** 10.3390/ijms23147983

**Published:** 2022-07-20

**Authors:** Katie L. J. Cederberg, Umaer Hanif, Vicente Peris Sempere, Julien Hédou, Eileen B. Leary, Logan D. Schneider, Ling Lin, Jing Zhang, Anne M. Morse, Adam Blackman, Paula K. Schweitzer, Suresh Kotagal, Richard Bogan, Clete A. Kushida, Yo-El S. Ju, Nayia Petousi, Chris D. Turnbull, Emmanuel Mignot

**Affiliations:** 1Department of Psychiatry and Behavioral Sciences, Stanford University, 3165 Porter Drive, Stanford, CA 94304, USA; kcederb@stanford.edu (K.L.J.C.); umaerhanif@hotmail.com (U.H.); vsempereperis@gmail.com (V.P.S.); jhedou@stanford.edu (J.H.); eileen@eileenleary.com (E.B.L.); logands@gmail.com (L.D.S.); linglin2058@gmail.com (L.L.); janezh@stanford.edu (J.Z.); clete@stanford.edu (C.A.K.); 2Biomedical Signal Processing & AI Research Group, Department of Health Technology, Technical University of Denmark, 2800 Kongens Lyngby, Denmark; 3Danish Center for Sleep Medicine, Department of Clinical Neurophysiology, 2600 Glostrup, Denmark; 4Jazz Pharmaceuticals, 3170 Porter Drive, Palo Alto, CA 94304, USA; 5Alphabet, Inc., 1600 Amphitheater Parkway Mountain View, Palo Alto, CA 94043, USA; 6Stanford/VA Alzheimer’s Research Center, 3801 Miranda Ave, Building 4, C-141, Mail Code 116F-PAD, Palo Alto, CA 94304, USA; 7Division of Child Neurology and Pediatric Sleep Medicine, Geisinger, Janet Weis Children’s Hospital, 100 N Academy Ave, Danville, PA 17822, USA; amorse@geisinger.edu; 8Department of Psychiatry, University of Toronto, Toronto, ON M5G 1L5, Canada; adam@mettascience.com; 9Sleep Medicine & Research Center, St. Lukes Hospital, 232 S. Woods Mill Road, Chesterfield, MO 63017, USA; paula.schweitzer@stlukes-stl.com; 10Department of Neurology, Mayo Clinic, 200 First St., Rochester, MN 55905, USA; kotagal.suresh@mayo.edu; 11College of Medicine, Medical University of South Carolina, 171 Ashley Ave, Charleston, SC 29425, USA; richard.bogan@bogansleep.com; 12Department of Neurology, Washington University, St. Louis, MO 63110, USA; juy@wustl.edu; 13Hope Center for Neurological Disorders, Washington University, St. Louis, MO 63110, USA; 14Center on Biological Rhythms and Sleep (COBRAS), Washington University, 1600 S. Brentwood Blvd, St. Louis, MO 63144, USA; 15Experimental Medicine Division, Nuffield Department of Medicine, University of Oxford, Headley Way, Headington, Oxford OX3 9DU, UK; nayia.petousi@dpag.ox.ac.uk; 16National Institute for Health Research Oxford Biomedical Research Centre, University of Oxford, Headley Way, Headington, Oxford OX3 9DU, UK; christopher.turnbull@ouh.nhs.uk; 17Oxford Centre for Respiratory Medicine, Oxford University Hospitals NHS Foundation Trust, Headley Way, Headington, Oxford OX3 9DU, UK

**Keywords:** obstructive sleep apnea, proteomics, apnea–hypopnea index, biomarkers, machine learning, treatment

## Abstract

Obstructive sleep apnea (OSA), a disease associated with excessive sleepiness and increased cardiovascular risk, affects an estimated 1 billion people worldwide. The present study examined proteomic biomarkers indicative of presence, severity, and treatment response in OSA. Participants (*n* = 1391) of the Stanford Technology Analytics and Genomics in Sleep study had blood collected and completed an overnight polysomnography for scoring the apnea–hypopnea index (AHI). A highly multiplexed aptamer-based array (SomaScan) was used to quantify 5000 proteins in all plasma samples. Two separate intervention-based cohorts with sleep apnea (*n* = 41) provided samples pre- and post-continuous/positive airway pressure (CPAP/PAP). Multivariate analyses identified 84 proteins (47 positively, 37 negatively) associated with AHI after correction for multiple testing. Of the top 15 features from a machine learning classifier for AHI ≥ 15 vs. AHI < 15 (Area Under the Curve (AUC) = 0.74), 8 were significant markers of both AHI and OSA from multivariate analyses. Exploration of pre- and post-intervention analysis identified 5 of the 84 proteins to be significantly decreased following CPAP/PAP treatment, with pathways involving endothelial function, blood coagulation, and inflammatory response. The present study identified PAI-1, tPA, and sE-Selectin as key biomarkers and suggests that endothelial dysfunction and increased coagulopathy are important consequences of OSA, which may explain the association with cardiovascular disease and stroke.

## 1. Introduction

Obstructive sleep apnea (OSA) is characterized by recurrent partial or complete obstructive events of the upper airway, resulting in arousals and oxygen desaturations [1]. Nocturnal polysomnography (PSG) is the current gold standard for diagnosing and evaluating sleep apnea severity [2]. The presence and severity of sleep apnea is approximated by the apnea–hypopnea index (AHI), the number of abnormal respiratory events recorded per hour of sleep during PSG, whereby an AHI ≥ 5 but <15 is indicative of mild sleep apnea, AHI ≥ 15 but <30 is indicative of moderate sleep apnea, and AHI ≥ 30 is indicative of severe sleep apnea in adults [3].

Obstructive sleep apnea is an extremely common condition. A recent study estimated almost 1 billion people worldwide are affected by OSA, with prevalence exceeding 50% in some countries [4]. The prevalence of OSA is higher in men, increases with age, and is associated with higher body mass index (BMI) [5]. A recent study in adults older than 40 reported the prevalence of moderate-to-severe sleep-disordered breathing was 49.7% in men and 23.4% in women [6]. High levels of sleep apnea, typically of the obstructive type, are associated with increased cardiovascular risk, notably increased risk of high blood pressure [7], increased vascular aging [8], heart failure [9], arrythmias [10], and stroke [11], effects that seem more correlated with hypoxia than with sleep fragmentation [10,12]. Sleep fragmentation, in contrast, correlates more with arousal disturbances and daytime sleepiness, although some authors have found higher cardiovascular risk in sleepy versus non sleepy OSA patients [10,13]. Timely diagnosis and treatment of OSA is imperative to the health and prognosis of millions.

Recent work highlighted a need for blood-based biomarkers in sleep medicine to streamline more accurate and objective ascertainment of disorders [14]. Previously, we studied 1300 serum proteins in 351 patients with OSA compared to 362 controls [15], identifying 65 proteins associated with AHI and 9 differentially expressed proteins (DEPs) in moderate-to-severe OSA (AHI ≥ 15) compared to mild OSA/controls (AHI < 15). That study demonstrated that multivariate protein measurement could be used to predict the presence of moderate-to-severe OSA (AHI ≥ 15) compared with mild OSA/controls (AHI < 15) using a machine learning classifier and identified 3 proteins that were responsive to positive airway pressure (PAP) treatment in a cohort of 16 patients with OSA. Such results supported the initial efficacy of proteomic biomarkers for diagnosing OSA as well as their potential for measuring OSA severity and treatment response.

In the present study, we extend and validate the results of our previous findings [15] by identifying proteins cross-sectionally associated with the presence and severity of OSA (i.e., total AHI and OSA severity classifications) using an independent sample with a larger panel (~5000) of proteins in a larger (~1400) sample of clinic patients. We also examined the effect of OSA treatment (i.e., CPAP/PAP and supplemental oxygen versus off treatment) on identified proteins to explore potential pathophysiological mechanisms associated with managing symptoms and consequences of OSA.

## 2. Results

### 2.1. Stanford Technology Analytics and Genomics in Sleep (STAGES) Study Cohort

Demographic and clinical information for participants (*n* = 1391) included in the final analyses is presented in Table 1. The final sample with all relevant outcomes had almost equal representation of males (52%) and females (48%), with an average age of 46.1 ± 15.2 years, average BMI of 30.9 ± 8.7 kg/m^2^, and was primarily Caucasian/white (80%) and non-Hispanic (94%). Regarding PSG, average sleep duration was 5.6 ± 1.7 h, and patients had an average AHI of 15.6 ± 19.7 (range = 0.0–154.6). According to PSG technician notes, 7% of participants were provided CPAP during the PSG. Approximately 7% of participants self-reported using CPAP or PAP, less than 1% reported supplemental oxygen use, and less than 1% reported having previous sleep apnea surgery. Regarding blood sampling, most samples (67%) were collected between 12:00 PM and 23:59 PM, with an average time from draw to blood processing of 31.0 ± 14.0 h (range = 6.8–126.3 h), during which blood was transported to Stanford University for central processing.

### 2.2. Effect of Time to Blood Processing

To analyze the effect of travel time at room temperature, we added 224 EDTA samples collected during the daytime that were immediately processed and were also measured using the same proteomic panel. Summary results of multivariate analyses of the combined cohort (*n* = 1654) including all STAGES plasma samples (*n* = 1430; time to processing range: 1–126 h) and the immediately processed samples (*n* = 224; time to processing = 0) are presented in Appendix A. Not surprisingly, most proteins (*n* = 4151) were affected by time from blood draw to processing, including 2505 positively associated proteins and 1646 negatively associated proteins, whereby relative protein expression increased or decreased, respectively, with longer time spent at room temperature. As this may be an important consideration in the interpretation of our results, time to blood processing was included as a covariate in all multivariate analyses. Further, we include false discovery rate (FDR) corrected *p*-values (i.e., FDR *p* combined time to processing) and beta coefficients (i.e., beta combined time to processing) from the analysis, including a comparison with immediately processed samples in Appendix A alongside results of the following multivariate analyses for additional consideration in the interpretation of our results.

### 2.3. Proteins Associated with the Apnea–Hypopnea Index

Proteins significantly associated with total AHI as a continuous variable are presented in Table 2 and Appendix A. Multivariate analyses identified 84 proteins (47 positively and 37 negatively) associated with AHI after FDR correction. Proteins most positively associated with AHI were E-selectin (sE-Selectin), tissue-type plasminogen activator (tPA), plasminogen activator inhibitor 1 (PAI-1), scavenger receptor cysteine rich type 1 protein (sCD163), and tissue factor pathway inhibitor (TFPI). Pathways related to fatty acid oxidation and metabolism, RA and retinol biosynthesis and metabolism, complement and coagulation cascades, blood coagulation and clotting cascades, platelet amyloid precursor protein, and plasminogen activating cascades were involved (Appendix A). Proteins most negatively associated with AHI include sex hormone-binding globulin (SHBG), amyloid-like protein 1, Desmoglein-2, serum albumin, and secretogranin-3 (SCG3), with pathways involved in insulin-like growth factor signaling and regulation, ghrelin regulation of food intake and energy homeostasis, genes encoding structural ECM glycoproteins, and post-translational modification of synthesis of GPI-anchored proteins (Appendix A).

### 2.4. Differential Expression of Proteins Stratified by Severity Index of AHI

Differences in characteristics of participants (*n* = 1391) by OSA classification status based on AHI are presented in Appendix A. Participants with moderate-to-severe OSA (AHI ≥ 15) were significantly older (50.5 ± 13.1 vs. 43.9 ± 15.7; *p* = 4.22 × 10^−15^), had higher BMI (33.6 ± 9.1 vs. 29.5 ± 8.1; *p* = 8.23 × 10^−23^), and consisted of a significantly higher proportion of males (63% vs. 41%; *p* = 1.63 × 10^−15^) compared with the mild-to-no OSA (AHI < 15) group (Appendix A). A marginally longer time to processing was found in subjects with moderate-to-severe OSA (32.1 ± 14.4 vs. 30.5 ± 13.8, respectively; *p* = 0.004). Multivariate analyses identified 69 differentially expressed proteins (DEPs) between these two OSA groups, with 40 upregulated and 29 downregulated in those with moderate-to-severe OSA (Appendix A). Top upregulated proteins included Quinone oxidoreductase-like protein 1 (QORL1), Tumor protein p53-inducible protein 11 (P5I11), cystine-rich protein 1 (CRIP1), DCN1-like protein 5 (DCNL5), and Tartrate-resistant acid phosphatase type 5 (TrATPase), with pathways similar to those listed for proteins positively associated with AHI (Appendix A). Top downregulated proteins included Amyloid-like protein 1, Desmoglein-2, SHBG, secretogranin-3 (SCG3), and vesicular, overexpressed in cancer, prosurvival protein 1 (ECOP), with pathways like those listed for proteins negatively associated with AHI, plus fibrinolysis pathways, complement system pathways, and others (Appendix A). In comparison to AHI-associated proteins found using linear regression, 50 of 69 DEPs for this OSA classification were significant in AHI multivariate analyses.

Summary characteristics comparing severe OSA, moderate OSA, mild OSA, and controls are presented in Appendix A. Thirty-two significant DEPs (20 upregulated, 12 downregulated) were found in severe OSA compared to controls, and 4 significant DEPs (1 upregulated, 3 downregulated) were found in moderate OSA compared to controls (presented in Appendix A, respectively). No DEPs were found between mild OSA and controls.

### 2.5. Protein Markers Validated with a Machine Learning Classifier

Many proteins (*n* = 84) were significantly associated with AHI when modeled as a continuous variable, and the largest number of proteins (*n* = 69) was identified when AHI was stratified by severity using the cutoff score of 15 as a binary variable. The AHI ≥ 15 cutoff score was therefore next used to train a machine learning classifier on SomaScan protein measures using three different models. Model 1 incorporated protein measures and demographic variables and achieved 66.4% accuracy (F1 = 0.598) in classifying moderate-to-severe OSA. Model 2 incorporated only proteins and achieved 66.9% accuracy (F1 = 0.603). Model 3, with only demographics, achieved the lowest accuracy of 63.5% (F1 = 0.566). Receiver operating curves (ROC) and areas under the curves (AUCs) for each model are presented in Figure 1, classifier metrics are presented in Table 3, and the confusion matrix is in Appendix A. Of the top 15 classifier features from Model 1 (Appendix A), 10 proteins were identified as significant protein markers of AHI, and 8 were significant DEPs for moderate-to-severe OSA (Amyloid-like protein 1, tPA, PAI-1, CRIP1, SCG3, ECOP, Desmoglein-2, and SHBG), suggesting these proteins are robust predictors of the presence and severity of sleep apnea. When comparing the models, we see a trend in the differences for Model 1 compared with Model 3, and Model 2 compared with Model 3 (Appendix A). It seems here that the improvement in prediction coming from the protein expression variables over the demographics is the main factor playing into the performance of the model.

### 2.6. Effect of CPAP and Supplemental Oxygen on Protein Markers

Proteins associated with CPAP/PAP intervention are presented in Appendix A. For this comparison, we only used the 84 proteins previously shown to be associated with total AHI. Nonparametric paired samples tests comparing pre- and post-intervention protein expression identified 5 proteins significantly reduced following use of CPAP/PAP: CRP (change = −12.3%; *p* = 0.004), PAI-1 (change = −10.6%; *p* = 0.007), TrATPase (change = −6.2%; *p* = 0.009), sE-Selectin (change = −8.2%; *p* = 0.030), and tPA (change = −14.1%; *p* = 0.047). These proteins are associated with pathways involved in blood coagulation, inflammatory response, fibrinolysis, and plasminogen activation. Regarding changes in protein expression with supplemental oxygen compared with air (i.e., sham), when patients were off CPAP, nonparametric paired comparisons identified significant differences in protein expression change for PAI-1 and TGF-b R III (Appendix A), whereby PAI-1 decreased with supplemental oxygen and increased with air, whereas TGF-b R III increased with supplemental oxygen and decreased with air. Regarding the proteins with significant changes following CPAP/PAP, CRP and tPA decreased, and TrATPase and sE-Selectin increased with supplemental oxygen, whereas these proteins all increased with air. However, these differences did not reach statistical significance.

### 2.7. Replication with Our Prior Study in Serum

In comparison with our previous findings in serum [15], 44 (52%) of the 84 proteins associated with total AHI in the present study were not included in the SomaScan 1.3 panel used in our previous study, and therefore could not be included in this comparison (see Appendix A for a complete list). Seven (18%) of the 40 remaining protein markers of AHI were replicated in the present study, including tPA, TFPI, Aminoacylase-1, LG3BP, Factor I, and Coagulation Factor IXab (all positively associated with AHI), and UNC5H4 (negatively associated with AHI). We further conducted multivariate linear regression analyses for the 1282 mutual proteins between the 1.3K and 5K panels in the combined STAGES/Stanford Sleep Study cohort (*n* = 1921). This analysis resulted in 77 proteins (49 positive, 28 negative) significantly associated with AHI (Appendix A), including 22 proteins identified in the STAGES cohort alone. Significant proteins included sE-Selectin, PAI-1, Coagulation Factor IXab, TFPI, IGFBP-2, UNC5H4, SHBG, CRP, TrATPase, and tPA, with pathway analysis identifying functions consistent with those previously described for AHI as a continuous variable (Appendix A).

We further trained a similar machine learning classifier as previously described with the 5 proteins significant for AHI in the STAGES cohort, combined cohort, and that were responsive to CPAP (i.e., CRP, PAI-I, TrATPase, tPA, and sE-Selectin) for classifying participants into OSA or control categories based on an AHI cutoff score of 15. ROCs and AUCs for each model are presented in Figure 2, and classifier metrics are presented in Table 4. Performance metrics for Model 1 (proteins + demographics) and Model 2 (proteins only) were similar to, albeit slightly lower than, those from the classifier using the entire array of 4985 proteins. Model 1 with the targeted proteins achieved the highest accuracy of 65.7% (F1 = 0.571) compared to an accuracy of 66.4% (F1 = 0.598) from the entire protein array. There were significant differences between Model 1 and Model 2 (DeLong: Z = 2.94; *p* = 0.003; Bootstrap: D = 2.95; *p* = 0.003) and Model 1 and Model 3 (DeLong: Z = 2.44; *p* = 0.014; Bootstrap: D = 2.43; *p* = 0.015; Appendix A). In this case, the integration of proteins with the demographics showed a clear improvement over the two individual data sources. Collectively, this suggests that this subset of proteins seems to contribute a significant proportion of the predictive ability of the presence, severity, and treatment response for obstructive sleep apnea.

## 3. Discussion

These results extend our prior study of 713 patients with 1500 proteins using an independent sample with a similar but larger platform and larger sample. In this study, we found 84 proteins (47 positive and 37 negatively) associated with total AHI, 50 of which were also significant DEPs between moderate-to-severe OSA (AHI ≥ 15) and mild/no OSA (AHI < 15). Of the 42 significant predictive features of the OSA machine learning classifier, 8 of the top 15 features were significant markers of total AHI and OSA from multivariate analysis, suggesting that these 8 proteins (Amyloid-like protein 1, tPA, PAI-1, CRIP1, SCG3, ECOP, Desmoglein-2, and SHBG) are likely predictors of the presence and severity of sleep apnea. Furthermore, five of the 84 significant protein markers for AHI (CRP, PAI-1, TrATPase, sE-Selectin, and tPA) were significantly affected by CPAP/PAP treatment and were replicated in a combined analysis with the samples from our previous study. These results suggest that this subset of proteins represents robust predictors of the presence and severity of sleep apnea and may be potential markers of treatment response in sleep apnea management.

The results obtained here must be considered more unbiased than candidate marker studies reported in the literature, as we studied over 5000 proteins, a study more akin to conducting a genome-wide association study versus candidate gene studies. Encouragingly, the present study validated markers identified in our previous findings with 1500 proteins in serum [15], notably increased tPA and sE-selectin, two factors that have been consistently reported in multiple studies to be associated with sleep apnea [16,17,18]. This is despite the fact the assays were done in EDTA plasma that had travelled for over 24 h at room temperature prior to processing in the STAGES samples. These two factors are primarily secreted by the endothelium in the setting of damage or injury, confirming that sleep apnea likely causes sleep fragmentation and damages the endothelium, perhaps as the result of hypoxia. These results support the notion that sleep apnea is associated with accelerated vascular aging [8]. Similarly, CRP, a marker of inflammation secreted by the liver and known to correlate with BMI, has been shown by multiple investigators to be higher in BMI-matched individuals with sleep apnea, and to also decrease with CPAP therapy [19,20].

A novel, potentially robust sleep apnea marker identified in this study is Tartrate-resistant acid phosphatase (TrATPase), also called acid phosphatase 5, tartrate resistant (ACP5). TrATPase is a glycosylated monomeric metalloprotein enzyme important to the function of osteoclasts, although it is also secreted by macrophages and immune cells. Deficiency in TrATPase is associated with spondylenchondrodysplasia, a disorder that combines abnormal bone development, various autoimmune diseases and central nervous abnormalities. Interestingly, OSA has been associated with osteoporosis in recent studies [21,22,23], a possible reflection of hypoxia.

The present study also identified PAI-1, another important coagulation factor, as significantly upregulated in sleep apnea, consistent with other studies [16,24]. Intriguingly, PAI-1 was not altered in our first published proteomic study, which used serum samples [15]. One possible explanation is that PAI-1 has a large platelet pool [25,26], thus measures in serum and plasma have different roles, explaining the discrepancy across the two studies, and suggesting that free PAI-1, the active form, is the culprit in the present findings. Another explanation could be that our first study was not powered to detect such an association in PAI-1, which would explain why PAI-1 was amongst the top proteins significantly associated with total AHI in the combined analysis.

In the context of pathobiological mechanisms underlying sleep apnea, PAI-1 is hypoxically regulated and an inhibitor of tPA, and dual elevation of PAI-1and tPA is associated with increases in coagulation and fibrinolysis turnover. Indeed, following injury, PAI-1 helps maintain fibrinolysis to local regions of tPA release [26,27]. Increased tPA and PAI-1 have also been suggested to increase local blood–brain barrier permeability [26]. More recently, increased PAI-1 and tPA have also demonstrated associations with post COVID-19 hypercoagulability [28,29], a state that induces thrombosis [30], a likely result of hypoxia. It is our hypothesis (Figure 3) that endothelial damage in sleep apnea, perhaps as the result of hypoxia, induces increased local fibrin aggregation and fibrinolysis. In this context, a dual increase of tPA and PAI-1 has been found to predict stroke [31], and they are further elevated post stroke [32,33], suggesting these to be possible mediators of the important association of stroke with sleep apnea. This may also explain the protective effect of CPAP in patients with OSA [34,35], whereby the present study identified significant reductions in PAI-1 and tPA, along with sE-Selectin, TrATPase, and CRP, after at least two weeks on CPAP/PAP. This suggests that treatment with CPAP/PAP reduces the hypoxic burden associated with sleep apnea, thereby reducing fibrinolysis, blood coagulation, plasminogen activation, and inflammatory responses.

There are important limitations to consider when interpreting our results. Although the SomaScan assay has demonstrated good validity and reproducibility as well as stability in protein expression with varying time to processing [36,37], the blood samples from different sites were shipped to Stanford for processing at room temperature, and therefore may have been subject to blood composition changes associated with such a delay. All PSGs were conducted as part of standard-of-care according to the clinic protocol at each site and using the sampling rate/filters recommended by the datacenter; therefore, certain methodologies may have varied between study sites. Similarly, evaluation of the response to CPAP/PAP treatment was conducted in a combined sample of participants from two separate cohorts with different methodologies. Neither of the intervention cohorts included a control condition. Furthermore, the combined sample size of 25 in CPAP/PAP evaluation may not be sufficiently powered to examine such a relationship, and future research should include larger sample sizes and comparisons with controls. We relied on self-reported outcomes for demographic and secondary clinical conditions (e.g., BMI, cardiovascular disease). Finally, although we found biomarkers that were cross-sectionally associated with sleep apnea and that responded to CPAP, our machine learning models achieved relatively low accuracy, and addition of these biomarkers to demographics only marginally improved performance in our ROC curve analysis. One reason for this could be the dependance of these markers on demographics, and/or that these markers only contribute marginally to risk of OSA. This result outlines the need for finding additional biomarkers by increasing the CPAP sample size.

## 4. Materials and Methods

### 4.1. Stanford Technology Analytics and Genomics in Sleep Study Cohort

The Stanford Technology Analytics and Genomics in Sleep (STAGES) study, described previously [38,39], collected data from patients across 11 different sleep clinics between 2018 and 2020. Briefly, all participants were patients who attended an appointment with a physician at a sleep clinic and completed an overnight polysomnography (PSG) study, in addition to completing the Alliance Sleep Questionnaire (ASQ) and providing a blood sample. Although 1430 participants completed the study, a total of 1391 participants had relevant PSG, ASQ, and blood sampling data and were included in final analyses. All investigations were carried out following the rules of the Declaration of Helsinki of 1975, each institution’s Institutional Review Board approved all study procedures, and participants provided written informed consent. Data collected from STAGES, aside from the blood samples, are available through the National Sleep Research Resource (NSRR).

#### 4.1.1. Polysomnography

All participants completed a Level 1 nocturnal PSG study in a sleep lab. All PSGs were conducted as part of standard-of-care according to the clinic protocol at each site and using the sampling rate/filters recommended by the datacenter. Outcomes of interest for the present study included total PSG duration (h), sleep duration (h), apnea–hypopnea index (AHI, with hypopnea defined as 3% desaturation or arousal). Individuals undergoing PSG exclusively for treatment purposes (e.g., oral appliance evaluation) were excluded at enrollment, but CPAP or other OSA treatment was used in some cases, and these participants were included in this analysis. AHI was scored based on the American Academy of Sleep Medicine’s recommended criteria for scoring a respiratory event as a hypopnea [3], and the presence and severity of OSA was characterized by the AHI score. A cutoff score of 15 was used to stratify patients to moderate-to-severe OSA (AHI ≥ 15) or mild-to-no OSA (AHI < 15). We also subclassified OSA cases into severe (AHI ≥ 30), moderate (30 > AHI ≥ 15), or mild (15 > AHI ≥ 5) in comparison to controls (AHI < 5) using standard cutoff scores [1,3].

#### 4.1.2. Alliance Sleep Questionnaire

The ASQ is an electronic, comprehensive sleep disorder questionnaire that includes questions and validated measures designed to collect standardized subjective sleep data [39,40]. The questionnaire uses complex branching logic in a modular fashion to guide participants through a comprehensive set of questions to quantify demographic and clinical characteristics, including age, sex, body mass index (BMI), race, ethnicity, and associated medical history (e.g., medication use, presence of comorbidities, previous diagnosis of sleep conditions, sleep complaints). All patients completed the ASQ within approximately 4 weeks of the polysomnography.

#### 4.1.3. Blood Sampling

Blood samples were typically collected during the scheduled PSG, either the evening before or the morning after the PSG, or within four weeks of the PSG. Times of blood draw were dichotomized (i.e., blood draw) into morning (i.e., 0:00–11:59) and evening (i.e., 12:00–23:59) samples to control for diurnal variation in blood composition. Blood samples were then shipped at room temperature, in most cases taking 1–2 days to arrive at Stanford, and processed for EDTA plasma and DNA extraction. As there was variability in the amount of time samples remained at room temperature (i.e., time to blood processing), time to processing was included in all analyses as a salient covariate. Time to processing was calculated as the total number of hours between time of blood sample collection to time plasma was frozen, which was then log-normalized, and extreme outliers (log-normalized processing time < 2.7) were removed (*n* = 7) to reduce variability in protein expression based on a long duration of processing time.

### 4.2. Intervention Cohorts

EDTA plasma samples from two different studies were used to examine the effect of CPAP/PAP and oxygen use on protein biomarkers of sleep apnea. One study recruited local community members and was conducted at Washington University in St. Louis (WashU) as described previously [41]. It included 16 patients with OSA (7 mild (5 < AHI < 15); 11 moderate-to-severe (AHI ≥ 15)) who underwent PAP treatment for 1–4 months and provided plasma samples pre- and post-treatment. Participants included in analyses were adherent to PAP, defined as usage for at least 4 h on at least 70% of the 30 preceding nights as recorded by the PAP machine. The other study recruited participants with a diagnosis of moderate-to-severe OSA who had been treated with CPAP for more than one year; this study was conducted at the University of Oxford, as described previously [42]. It included 25 participants in a two-arm randomized crossover design. Participants had moderate-to-severe OSA and had been treated with CPAP for more than 1 year, with average CPAP usage exceeding 4 h/night. After four nights of screening off CPAP, participants had at least 14 nights back on CPAP therapy before undergoing 14 nights of supplemental oxygen or 14 nights of air (both off CPAP). Participants then had a washout period of at least 14 nights back on CPAP before crossing over (e.g., air if oxygen was the first arm). Plasma samples were collected on days 0 (pre-arm 1), 14 (post-arm 1), 28 (pre-arm 2), and 42 (post-arm 2). The present study combined 16 pre-PAP samples (from the WashU study) with 25 post-air off-CPAP samples (from the Oxford study) as the “before CPAP/PAP” condition and 16 post-PAP samples (WashU) with 25 pre-air on-CPAP samples (Oxford) as the “after CPAP/PAP” condition, for a total sample size of 41 participants. We further examined the effect of supplemental oxygen compared with that of air from the samples collected (*n* = 25) in the Oxford study.

### 4.3. Stanford Sleep Cohort

We sought to extend and replicate our previous findings using the Stanford Sleep Study, as described previously [15]. Briefly, 1070 participants ages 18 to 91 were enrolled by the Stanford Sleep Clinic in 1999, and each participant provided a blood sample and completed overnight PSG. Blood samples were typically collected the morning following overnight PSG and allowed to clot for a minimum of 30 min. Serum was then aliquoted and stored at −80 °C before being shipped to SomaLogic for protein quantification. Of note, serum samples from this study were collected and processed with a mean delay of 11.6 years (i.e., from blood draw to SomaScan assay). PSG studies for cohort participants were scored using the alternate AASM hypopnea definition for AHI [43].

### 4.4. Protein Quantification

Plasma samples from the STAGES cohort and intervention cohorts and serum samples from the Stanford Sleep Cohort followed the same protein quantification protocol. Relative expression levels (i.e., relative florescent units (RFU)) of proteins were assayed using the SomaScan aptamer-based multiplexed platform (SomaLogic Inc., Boulder, CO, USA), which utilizes aptamers and hybridization to quantify proteins from small amounts of human plasma [44,45,46]. The SomaScan platform was designed to have extended dynamic range and includes both extracellular and intracellular proteins (with soluble domains of membrane proteins). SomaScan assays have demonstrated good validity and reproducibility as well as stability in protein expression with varying time to processing [36,37]. SomaLogic further conducts standardized data quality control at the sample level and protein level to adjust for variability between and within samples, and provides population-based normalized outputs of relative protein expression levels. Detailed information on SomaLogic’s quality control technique can be found on the manufacturer’s website (https://somalogic.com/technology/, accessed on 17 July 2022). SomaLogic provided two output files, each with different levels of population-based normalization, whereby the present study utilized the most-normalized output based on our sample distributions. Plasma samples from the STAGES cohort and intervention cohorts used the SomaScan platform of 5287 proteins (i.e., version 5K). After removal of non-human proteins (e.g., mouse), a total of 4985 proteins were quantified and analyzed (see Appendix A for a complete list of proteins included in analyses). Serum samples from the Stanford Sleep Cohort used the SomaScan platform of 1300 proteins (i.e., version 1.3K).

### 4.5. GWAS Principal Components

As genetic ancestry plays an integral role in disease phenotyping, the first five principal components from genome-wide association study (GWAS) analysis were included as important covariates in the present study. A total of 467 patients with OSA and 907 controls, as defined above, all genotyped using an Affymetrix PMRA array (Thermo Fisher Scientific, Santa Clara, CA, USA), were included. The cohort was imputed to the 1000 Genome Phase III [47] after haplotype phasing using QCTOOL version 2. Quality control was conducted to ensure no duplicated patients based on inheritance by descent using KING version 2.2.6 [48]. Variants and samples with missing call rates lower than 0.1, and variants with minimum allele frequency lower than 0.05 were excluded. Imputation calls with *R*^2^ ≥ 0.9 were selected to compute principal component analysis using PLINK version 1.9 [49,50]. Patients were matched with the closest controls in a 1:2 ratio based on the Euclidean distance of individuals’ principal components.

### 4.6. Statistical Analyses

#### 4.6.1. Descriptive Statistics

Data were analyzed in Jupyter Notebook version 6.0.1 with Python 3.7.4 using the statsmodels library version 0.10.1 [51]. Descriptive statistics are reported as mean and standard deviation (SD) for continuous variables, and number and percentage for dichotomous variables. We examined differences in demographic and clinical characteristics between participants with OSA and controls using Mann–Whitney U tests, as most of our continuous variables were not normally distributed, and Chi squared tests for dichotomous variables, whereby an a priori alpha level of 0.05 indicated a significant difference between groups.

#### 4.6.2. Multivariate Proteomic Analyses

Using the statsmodels 0.10.1 library in Python 3.7.4 [51], we executed a series of multiple linear regression models with log-normalized protein expression as the dependent variable and the associated sleep apnea feature as the independent variable. Important covariates such as age, sex, BMI, log-normalized blood time to processing, blood draw period (i.e., morning or evening), study site, and the first five principal components from GWAS analysis to control for ancestry were included. As BMI is highly correlated with sleep apnea, we sought to include additional features as covariates to control for any residual effect, including BMI^2^ and age × BMI × sex. We executed regression models with the following sleep apnea features as independent variables: (1) log-normalized AHI + 1; (2) mild-to-moderate OSA vs. mild/no OSA; (3) severe OSA vs. controls; (4) moderate OSA vs. controls; and (5) mild vs. controls. We applied the Benjamini–Hochberg procedure for controlling the FDR to all *p*-values with an a priori *p*-value of 0.05 for identifying statistically significant proteins in all models.

#### 4.6.3. Effect of Time to Blood Processing on Protein Expression

As our samples were collected at 11 different sites across the country, time to blood processing varied considerably. To account for this, we examined the effect of time to processing in STAGES samples (*n* = 1430; time to processing range: 1–126 h) in comparison to 224 samples processed immediately after collection. Multiple linear regressions with log-normalized protein expression as the dependent variable and log-normalized time to blood processing as the independent variable with a 5% FDR correction were used for this comparison. An FDR-corrected a priori *p*-value of 0.05 was used for significance. To facilitate interpretation of our findings, *p*-values and beta coefficients for the effect of time to processing are presented in the Appendix A alongside multivariate results for each sleep apnea feature performed in the STAGES dataset.

#### 4.6.4. Pathway Analysis

The ToppFun module of the ToppGene suite (Division of Biomedical Informatics, Cincinnati Children’s Hospital Medical Center (BMI CCHMC), Cincinnati, OH, USA) was utilized for biological pathway enrichment analysis for all 5% FDR significant protein sets [52]. The ToppGene Suite is a free web portal (http://toppgene.cchmc.org, accessed on 25 April 2022) that executes candidate gene/protein prioritization using functional annotations or network analysis. The ToppFun module detects functional enrichment of the input gene list based on gene ontology, protein domains and interactions, regulome, ontologies, phenotype, pharmacome, and bibliome, and applies FDR and Bonferroni correction for determining statistical significance. The goal of this network-based prioritization is to identify proteins/genes that are relevant to pathways involved in biological processes or diseases. The NCBI Entrez gene IDs of all 5% FDR significant proteins were entered into the ToppFun module, and results are presented based on significant pathways/functions identified by ToppFun for all FDR significant proteins as well as upregulated and downregulated significant proteins separately.

#### 4.6.5. Validation of Protein Markers with a Machine Learning Classifier

The Python Scikit-learn LogisticRegressionCV library (version 1.0.2) [53] was used to train an ElasticNet (L1 and L2) penalty regularized lasso model to classify participants as OSA cases (AHI ≥ 15) or mild-to-no OSA (AHI < 15) based on relative protein expression. Hyperparameters of the models were tuned via 5-fold cross-validation using RepeatedKFold model evaluation. We used a 70% train–30% test split, whereby data on 974 (331 cases, 643 controls) participants were used for model training, and the remaining 417 (142 cases, 275 controls) participants were used for model testing. Three models were trained to predict OSA, each with the following features: (1) protein measures in addition to demographic information (age, sex, and BMI); (2) protein measures alone; and (3) demographic information (age, sex, and BMI) alone. Preprocessing for the classifier included log-normalization of relative protein expression and time to blood processing, followed by the conversion of all continuous variables to a standardized scale using the sklearn.preprocessing.StandardScaler feature. All models and classifiers used the same preprocessing, test–train split, and elastic net-based approach for comparison. We used two strategies to compare the ROC curves with statistical tests: DeLong test and a bootstrap approach. The DeLong test is a generalization of the Mann–Whitney statistic approach for the comparison of ROC curves [54]. The null hypothesis is that both areas under the curve are equal. The nonparametric asymptotic behavior for U-statistics is valid and can generalize in this case. The bootstrap approach consists of resampling bootstrap replicates of the data and comparing the distribution of the differences of AUCs to the normal distribution.

#### 4.6.6. Effect of CPAP and Supplemental Oxygen Interventions on Sleep Apnea Proteins

Using the Python statsmodels library, Wilcoxon signed-rank sum tests, the nonparametric equivalent of a paired *t*-test, were conducted to examine if there were statistically significant differences in protein expression after CPAP/PAP compared to before CPAP/PAP in the combined WashU and Oxford cohort (*n* = 41). We further examined the effect of supplemental oxygen on proteins associated with AHI in comparison with air (i.e., sham) condition in the Oxford study (*n* = 25) by conducting Wilcoxon signed-rank sum tests between change scores in protein expression (i.e., post–pre values) after the oxygen condition and the air condition.

#### 4.6.7. Replication with Our Prior Study in Serum

Lastly, we sought to replicate and validate our findings with multivariate and classifier analyses in a combined cohort of the samples from the present STAGES study (*n* = 1391) and samples from our previous study (i.e., Stanford Sleep Study [15]) including 530 participants. As our previous study utilized the SomaScan 1.3K platform, these analyses were conducted in a total of 1282 proteins who were present in both the 1.3K and 5K panels used in the STAGES study. We conducted multiple linear regression analysis in the combined cohort (*n* = 1921), including log-normalized protein expression as the dependent variable, log-normalized AHI + 1 as the independent variable, and covariates such as age, sex, BMI, BMI^2^, age × sex × BMI, blood draw period, GWAS principal components, and study (i.e., STAGES or Stanford Sleep Study).

## 5. Conclusions

The present study conducted the largest examination of proteins, to date, in adults with OSA. Confirming our results, PAI-I and tPA were robust predictors based on multivariate analyses and the machine learning classifier, and were replicated in the combined analysis. Both were also responsive to CPAP/PAP treatment. Reductions in PAI-1 after both supplemental oxygen and CPAP/PAP, but increases with the sham condition are consistent with previous studies [16,24], and reductions in sE-Selectin after CPAP/PAP is consistent with results of a recent meta-analysis [55]. Collectively, these findings support the use of PAI-I, tPA, and sE-Selectin as biomarkers of the presence and severity of sleep apnea, and they may provide a measure of treatment response in people with sleep apnea. These results may also inform considerations for additional preventive cardiovascular management for high-risk patients. Measurement of these key biomarkers may add to the patient profile and potentially inform additional therapeutic strategies even outside CPAP/PAP use, and creates an opportunity to inform cardiovascular disease management.

## Figures and Tables

**Figure 1 ijms-23-07983-f001:**
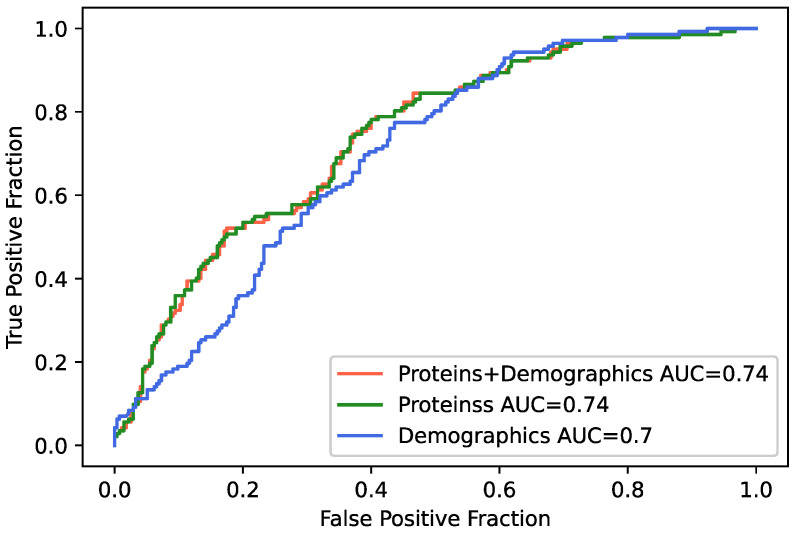
Receiver operating characteristics (ROC) curves for machine learning classifier for moderate-to-severe obstructive sleep apnea. Model 1 (red line) included all 4985 proteins as well as demographic variables (i.e., age, sex, BMI); Model 2 (green line) included only the 4985 proteins; and Model 3 (blue line) included only demographic variables.

**Figure 2 ijms-23-07983-f002:**
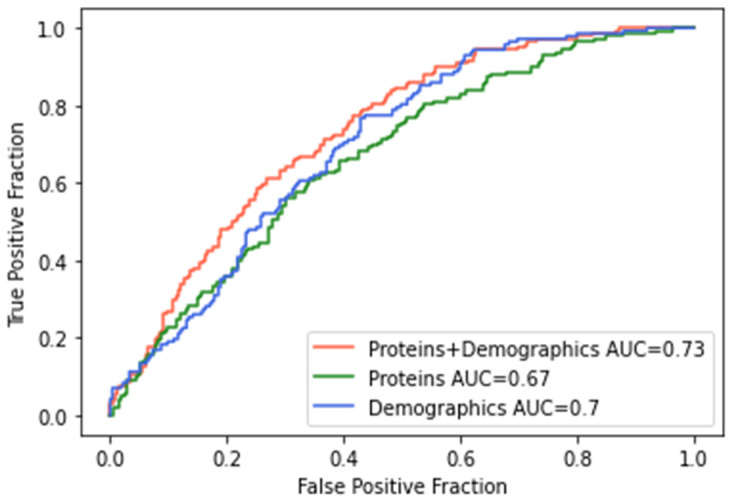
Receiver operating characteristics (ROC) curves for machine learning classifier for moderate-to-severe obstructive sleep apnea in replicated and CPAP proteins. Model 1 (red line) included the five replicated proteins as well as demographic variables (i.e., age, sex, BMI) and achieved the highest accuracy of 66%; Model 2 (green line) included only the five proteins, and Model 3 (blue line) included only demographic variables.

**Figure 3 ijms-23-07983-f003:**
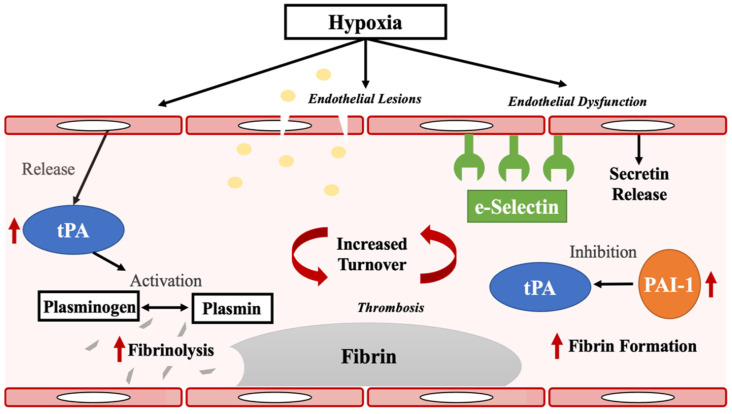
Proposed mechanisms in the relationship between obstructive sleep apnea, PAI-1, tPA, and sE-Selectin. Hypoxia in sleep apnea causes endothelial damage or injury. This is associated with (1) increased release of sE-selectin; and (2) stimulation of local fibrin aggregation. Increased tPA and PAI-1 activity reflects enhanced fibrin formation/fibrinolysis turnover, perhaps contributing to the known increased risk of stroke in sleep apnea.

**Table 1 ijms-23-07983-t001:** Demographic and clinical characteristics of participants included in final analyses (*n* = 1391).

	Mean	Std	Min	Max
Demographic Characteristics				
Age (years)	46.1	15.2	13.0	84.0
Sex (M)	720 (52%)			
BMI (kg/m^2^)	30.9	8.7	11.9	75.0
Race				
Caucasian/White	1112 (80%)			
Asian	141 (10%)			
Two or More	89 (6%)			
Other (Black/African American/American Indian/Alaska Native/Pacific Islander)	49 (4%)			
Ethnicity (Hispanic Origin; yes)	87 (6%)			
Comorbidities				
Hypertension	430 (32%)			
Cardiovascular Problems	175 (13%)			
High Cholesterol	373 (27%)			
Diabetes II	129 (9%)			
Asthma	283 (21%)			
Chronic Obstructive Pulmonary Disease	57 (4%)			
Other Pulmonary Problems	52 (3%)			
Polysomnography Outcomes				
Sleep Duration (h)	5.6	1.7	1.0	10.2
AHI (number/hour)	15.6	19.7	0.0	154.6
Total Apnea events (n)	30.1	60.7	0.0	862.0
Average Apnea Duration (s)	19.4	7.7	6.5	63.8
Total Hypopnea Events (n)	44.3	63.1	0.0	528.0
Average Hypopnea Duration (s)	20.8	7.9	0.0	71.8
Total Desaturations (n)	35.8	74.8	0.0	866.0
Average Desaturation Duration (s)	29.4	11.1	5.0	98.4
CPAP Use During PSG	94 (7%)			
Previous OSA Diagnosis	273 (20%)			
CPAP/PAP Use	98 (7%)			
Supplemental Oxygen Use	11 (<1%)			
Prior Sleep Apnea Surgery	8 (<1%)			
Plasma Sample Characteristics				
Draw Period (*n* (%))				
AM (6:00–11:59)	455 (33%)			
PM (12:00–23:59)	936 (67%)			
Time to Blood Processing (h)	31.0	14.0	6.8	126.3

Notes: Data are presented as number (%) unless otherwise specified: std, standard deviation; BMI, body mass index; OSA, obstructive sleep apnea; CPAP, continuous positive airway pressure; PAP, positive airway pressure; PSG, polysomnography; AHI, apnea–hypopnea index.

**Table 2 ijms-23-07983-t002:** The 84 proteins significantly associated with the apnea–hypopnea index (AHI) after 5% FDR correction.

Target	UniProt ID	Entrez Gene Symbol	FDR *p*	β
sE-Selectin	P16581	SELE	0.000029	0.06051
SHBG	P04278	SHBG	0.000029	−0.089355
Amyloid-like protein 1	P51693	APLP1	0.000029	−0.045214
Desmoglein-2	Q14126	DSG2	0.000029	−0.034139
tPA	P00750	PLAT	0.000031	0.069885
PAI-1	P05121	SERPINE1	0.000036	0.063664
Albumin	P02768	ALB	0.000062	−0.020515
SCG3	Q8WXD2	SCG3	0.000162	−0.033395
sCD163	Q86VB7	CD163	0.000205	0.042673
NEGR1.2	Q7Z3B1	NEGR1	0.000225	−0.018214
NEGR1.1	Q7Z3B1	NEGR1	0.000225	−0.018214
SEZ6L	Q9BYH1	SEZ6L	0.00033	−0.020777
TFPI	P10646	TFPI	0.00033	0.029951
Aminoacylase-1	Q03154	ACY1	0.000471	0.075644
P5I11	O14683	TP53I11	0.001001	0.085686
GP116	Q8IZF2	ADGRF5	0.001001	0.051286
NAR3	Q13508	ART3	0.001001	−0.033656
IGFBP-5	P24593	IGFBP5	0.001157	−0.036965
IGFBP-2	P18065	IGFBP2	0.001182	−0.056639
Agrin	O00468	AGRN	0.001399	0.030074
ADH4	P08319	ADH4	0.001399	0.078678
CRIP1	P50238	CRIP1	0.001419	0.042984
QORL1	O95825	CRYZL1	0.001419	0.059314
CPLX2	Q6PUV4	CPLX2	0.001419	−0.040126
TrATPase	P13686	ACP5	0.00142	0.028185
LG3BP	Q08380	LGALS3BP	0.001451	0.043542
TMCC3	Q9ULS5	TMCC3	0.001893	0.04806
Adiponectin	Q15848	ADIPOQ	0.00193	−0.048973
Retinal dehydrogenase 1	P00352	ALDH1A1	0.002368	0.050139
ECOP	Q96AW1	VOPP1	0.002368	−0.03208
RGMB	Q6NW40	RGMB	0.002787	−0.01764
IL-1F6	Q9UHA7	IL36A	0.004977	0.049501
MXRA8	Q9BRK3	MXRA8	0.005084	−0.021983
Apo F	Q13790	APOF	0.005695	−0.049634
DCNL5	Q9BTE7	DCUN1D5	0.005695	0.034994
TGF-b R III	Q03167	TGFBR3	0.006	−0.017724
DKK3	Q9UBP4	DKK3	0.006746	−0.022198
BGLR	P08236	GUSB	0.006973	0.05775
ALDOB	P05062	ALDOB	0.006973	0.060758
NG36	Q96KQ7	EHMT2	0.006973	−0.034322
GLTD2	A6NH11	GLTPD2	0.00784	0.025655
SSRA	P43307	SSR1	0.008692	0.023203
LSAMP	Q13449	LSAMP	0.009783	−0.016283
Nectin-like protein 3	Q8N3J6	CADM2	0.009832	−0.028616
ADH1G	P00326	ADH1C	0.010797	0.082389
Keratin 7	P08729	KRT7	0.011084	−0.035521
ADH1A	P07327	ADH1A	0.012978	0.048781
Notch-3	Q9UM47	NOTCH3	0.013526	−0.016603
WISP-2	O76076	WISP2	0.013526	−0.02948
ATF6B	Q99941	ATF6B	0.014183	0.021569
Siglec-7	Q9Y286	SIGLEC7	0.014183	0.019849
HTRA1	Q92743	HTRA1	0.014939	0.02271
GPDA	P21695	GPD1	0.015466	0.041435
LECT2	O14960	LECT2	0.015466	0.040275
UNC5H4	Q6UXZ4	UNC5D	0.015833	−0.040023
CNTFR alpha	P26992	CNTFR	0.016	−0.018317
CRP	P02741	CRP	0.016692	0.085334
TSG-6	P98066	TNFAIP6	0.017048	−0.036688
CRDL1	Q9BU40	CHRDL1	0.017249	−0.019389
EphB6	O15197	EPHB6	0.017249	−0.014918
SERC	Q9Y617	PSAT1	0.019394	0.059281
APEL	Q9ULZ1	APLN	0.019471	0.017664
Factor I	P05156	CFI	0.019664	0.010343
NOTUM	Q6P988	NOTUM	0.021415	0.031373
TICN3	Q9BQ16	SPOCK3	0.02557	0.04362
SDC3	O75056	SDC3	0.026477	0.024861
TPMT	P51580	TPMT	0.029611	0.024919
SLIK1	Q96PX8	SLITRK1	0.030448	−0.035068
Cathepsin A	P10619	CTSA	0.031241	0.037457
IGF-II receptor	P11717	IGF2R	0.036093	0.018009
DUSP13	Q6B8I1	DUSP13	0.036472	−0.032443
SCG1	P05060	CHGB	0.037039	−0.020791
Cytochrome P450 3A4.2	P08684	CYP3A4	0.03901	0.030195
Cytochrome P450 3A4.1	P08684	CYP3A4	0.03901	0.030195
HEM4	P10746	UROS	0.039722	0.027314
GGT2	P36268	GGT2	0.039722	0.047757
JTB	O76095	JTB	0.041075	−0.016475
TPP1	O14773	TPP1	0.042957	0.025119
CBPM	P14384	CPM	0.042957	0.021753
ARMEL	Q49AH0	CDNF	0.044989	−0.02417
AZGP1	P25311	AZGP1	0.044989	−0.014643
Coagulation factor IXab	P00740	F9	0.046407	0.021377
Macrophage mannose receptor	P22897	MRC1	0.046663	0.01932
PGD2 synthase	P41222	PTGDS	0.049592	−0.024343

**Table 3 ijms-23-07983-t003:** Performance metrics of a machine learning classifier for moderate-to-severe OSA (AHI ≥ 15) trained on 974 samples (331 cases, 643 controls) and validated on 417 samples (142 cases, 275 controls).

Model	F1	Accuracy	Sensitivity	Specificity	AUC
1. Proteins + Demographics	0.598	0.664	0.732	0.629	0.741
2. Proteins	0.603	0.669	0.739	0.633	0.741
3. Demographics	0.566	0.635	0.697	0.604	0.700

**Table 4 ijms-23-07983-t004:** Performance metrics of a machine learning classifier using only the 5 replicated proteins that were also CPAP responsive for moderate-to-severe OSA (AHI ≥ 15) trained on 974 samples (331 cases, 643 controls) and validated on 417 samples (142 cases, 275 controls).

All Proteins Model	F1	Accuracy	Sensitivity	Specificity	AUC
1. Proteins + Demographics	0.571	0.657	0.669	0.651	0.734
2. Proteins	0.534	0.619	0.641	0.607	0.670
3. Demographics	0.566	0.635	0.697	0.604	0.700

## Data Availability

Data presented in this study from the STAGES cohort study, aside from the blood samples, are openly available as part of the National Sleep Research Resource (NSRR) at https://sleepdata.org/datasets/stages (accessed on 17 July 2022), and blood data may be made available upon request from the corresponding author.

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
