# Peer review of "Proteomic Biomarkers of the Apnea Hypopnea Index and Obstructive Sleep Apnea: Insights into the Pathophysiology of Presence, Severity, and Treatment Response"

_ijms, 2022, doi:10.3390/ijms23147983_

Round 1
Reviewer 1 Report
Thank you for the oportunity to review the manuscript entitled ”Proteomic Biomarkers of the Apnea Hypopnea Index and Obstructive Sleep Apnea: Insights into the Pathophysiology of Presence, Severity, and Treatment Respons”.
The objectives of the research are interesting but the mansucript must be re-write and meet journal's requirements.
Please revise the Introdiuction section - enlarge the information.
Matherial and emthods sections, ethical concern, statistical analysis data, results and interpretation of results, Also, Conclusion section is mandatory.
It is written as a ~rapport~ style rather than scientific article style.
Author Response
Thank you for the review of our manuscript. We have addressed each of your critiques below as indicated in bold text for ease of identification.
The objectives of the research are interesting but the mansucript must be re-write and meet journal's requirements.
Thank you for your review of our manuscript. Although the IJMS Microsoft Word Template File and the Instructions for Authors (https://www.mdpi.com/journal/ijms/instructions) were followed closely in preparing this manuscript, we have revised or provided a rationale for not revising the following concerns.
Please revise the Introdiuction section - enlarge the information.
We did not enlarge the text of this section as the formatting guidelines were followed in preparing this section, including the required font style and size as illustrated in the Microsoft template.
Matherial and emthods sections, ethical concern, statistical analysis data, results and interpretation of results, Also, Conclusion section is mandatory.
The Materials and Methods section includes the methods, including data curation from each sample cohort and statistical analyses. We also include the interpretation of results within the Results and Discussion sections. Although the website and word template state “Conclusions: This section is not mandatory but can be added to the manuscript if the discussion is unusually long or complex”, we now include a Conclusion section following the Methods and Materials section. We had originally included the conclusion as the last paragraph of the Discussion section as we found this to be clearer and improve readability rather than including it as its own section after the Methods and Materials section.
It is written as a ~rapport~ style rather than scientific article style.
We followed the guidelines and required sections of a “Research Manuscript” as outlined in the instructions for authors as well as the word template for original research articles. We will be happy to address more specific formatting concerns if necessary.
Reviewer 2 Report
The manuscript represents an extension of an earlier study on the relationship between OSA and proteins. I have some considerations.
"Previously, we studied 1300 serum proteins in 351 OSA patients compared to 362 controls" if this is an extension of the previous article which has 713 subjects, why does this manuscript have 1391? Please explain.
On the other hand, in the discussion section you write "700 participants" seems a bit contradictory.
The methodological section should contain more information about the procedure. I understand that it is a continuation of another study, but the reader should be able to know the procedure of this manuscript without reading another previous article. Who collected the sample? How were the participants contacted? What differences were there between the experimental and control groups? What was the protocol for protein analysis? Why were these proteins chosen?
This information could be for a better understanding of the results.
The results are very well explained with a lot of information, also the tables are clearly explained.
Thanks to the authors for explaining the limitations of the study.
Author Response
The manuscript represents an extension of an earlier study on the relationship between OSA and proteins. I have some considerations.
Thank you for the review of our manuscript. We have addressed each of your critiques below as indicated in bold text for ease of identification.
"Previously, we studied 1300 serum proteins in 351 OSA patients compared to 362 controls" if this is an extension of the previous article which has 713 subjects, why does this manuscript have 1391? Please explain.
Although we sought to extend our previous findings, this manuscript included an independent replication in a different sample (N=1391) and using a larger number of proteins (~5,000). We further replicated the findings of our prior study by utilizing a combined analysis of data from our previous study in 713 participants in addition to the present sample of 1391 with 1282 mutually present proteins. This is now clarified in the Introduction and Discussion sections.
On the other hand, in the discussion section you write "700 participants" seems a bit contradictory.
Thank you for this important observation. We have revised the Discussion section to be consistent with the aforementioned 713 participants.
The methodological section should contain more information about the procedure. I understand that it is a continuation of another study, but the reader should be able to know the procedure of this manuscript without reading another previous article. Who collected the sample? How were the participants contacted? What differences were there between the experimental and control groups? What was the protocol for protein analysis? Why were these proteins chosen? This information could be for a better understanding of the results.
As this manuscript utilized sample data from several sources, we reviewed each source separately for ease of identification.
1. The main sample is described in 4.1 Stanford Technology Analytics and Genomics in Sleep Study Cohort and includes a brief overview of the procedure for that study (e.g., samples were collected across 11 different sleep clinics and participants were patients of the sleep clinic that attended an appointment with a physician at a sleep clinic). Differences between groups based on OSA classifications are briefly described in section 2.4 Differential Expression of Proteins Stratified by Severity Index of AHI and represented in Supplementary Table S4 and this is now clarified in section 2.4 of the Results.
2. The intervention cohorts are described in 4.2 Intervention Cohorts and each cohort description includes a brief overview of who/where samples were collected and how participants were recruited (e.g., local community members and patients with a diagnosis of OSA). Neither of the intervention cohorts included a control condition for comparison and this is now included in our limitations.
3. The original study cohort is now described in greater detail and included as section 4.3 Stanford Sleep Cohort including the requested information.
Regarding protein analysis, the protocol for protein analysis is outlined in 4.4 Protein Quantification. Samples from the STAGES cohort and intervention cohorts followed the same protein quantification protocol and this is now clarified in section 4.4. The present study did not specifically choose or select proteins for analysis and included all human proteins represented in the SomaScan assay as the largest array of proteins available to measure in human plasma. Only non-human proteins were excluded from final analyses for being irrelevant to the present study and this is specified in section 4.4.
The results are very well explained with a lot of information, also the tables are clearly explained. Thanks to the authors for explaining the limitations of the study.
Thank you for the overall positive review of our manuscript.
Reviewer 3 Report
This manuscript describes proteomic biomarkers of apnea hypopnea index and obstructive sleep apnea through cohort study.
This is a very well-performed study and generates abundant information on the apnea hypopnea.
However there are several points to be addressed.
PAI-1 and tPA should be described in the abstract.
Fig1-2 legends should be described.
Fig3 the relationship between obstructive sleep apnea and hypoxia should be described.
Author Response
This manuscript describes proteomic biomarkers of apnea hypopnea index and obstructive sleep apnea through cohort study. This is a very well-performed study and generates abundant information on the apnea hypopnea. However there are several points to be addressed.
Thank you for the overall positive review of our manuscript. We have addressed each of your critiques below as indicated in bold text for ease of identification.
PAI-1 and tPA should be described in the abstract.
Thank you for this suggestion. The abstract now highlights PAI-1 and tPA as key biomarkers of OSA in the present study.
Fig1-2 legends should be described. Fig3 the relationship between obstructive sleep apnea and hypoxia should be described.
We now include figure legends for Figures 1-3.
Round 2
Reviewer 2 Report
,
Thank you very much for your explanations that help me to better understand the manuscript.
The figures are better explained.
Good job